

# The relationship between the living environment and remote working: an analysis using the SHEL model

Toshihisa Doi

Department of Living Environment Design, Osaka Metropolitan University, Osaka, Japan

## ABSTRACT

**Objective**. This study investigates the relationship of living environment factors with satisfaction, work engagement, perceived productivity, and stress among teleworkers.
**Background**. Given the increase in telecommuting since the onset of the pandemic, the study aims to identify how to create an optimum environment for telecommuting workers.
**Methods**. By examining the relationships among these factors *via* multiple regression analysis, a comprehensive investigation of the telecommuting working environment is conducted, encompassing physical aspects and facilities as well as lifestyle rhythms and relationships with housemates. In doing so, the author identifies measures to create a more favorable living environment. The work environment of remote workers is examined from various perspectives using the framework of the SHEL model: Software (work content, lifestyle, *etc.*), Hardware (furniture, equipment, *etc.*), Environment (indoor environment), and Liveware (relationships with family members who reside with the worker).
**Results**. The results suggest that positive factors, such as satisfaction and work engagement, are strongly influenced by the degree of job autonomy and the availability of a workspace dedicated to personal use. Negative aspects, such as stress, are significantly impacted by environmental noise, interruptions due to household tasks, and the use of ergonomic furniture.

Corresponding author
Toshihisa Doi, tdoi@omu.ac.jp

# INTRODUCTION

The coronavirus outbreak in 2019 changed how people work and the work environment (*Douglas et al., 2020*; *Wenham, Smith & Morgan, 2020*; *Belzunegui-Eraso & Erro-Garcés, 2020*). The prevalence of telecommuting rose significantly following the onset of the pandemic (*Eurofound and International Labour Office, 2017*; *Henke et al., 2016*), and the number of telecommuters increased significantly (*Ministry of Internal Affairs and Communications of Japan, 2023*). Telecommuting is thought to offer multiple advantages, including improved integration of home and work life, reduced fatigue, and increased productivity (*Gajendran & Harrison, 2007*). Many studies suggest that it may be preferable for workers to be situated in work from home in terms of well-being, quality of life, and

the reduction of stress and fatigue (*Anderson, Kaplan & Vega, 2015*; *Bosua et al., 2013*; *Fílardí, de Castro & Zaníní, 2020*; *Hayman, 2010*; *Kim et al., 2020*; *Tietze & Nadin, 2011*; *Tustin, 2014*). *Evans, Kunda & Barley (2004)* state that work-from-home arrangements can increase time flexibility by reducing commuting time. Even as the restrictions of the pandemic eased, many companies continued to permit telecommuting due to the anticipated benefits.

Studies by *Song & Gao (2020)*, *Windeler, Chudoba & Sundrup (2017)*, and *Mann & Holdsworth (2003)* report that telecommuting increases stress and fatigue. *Allen, Golden & Shockley (2015)* find that the blurring of physical and organizational boundaries between work and home can negatively impact the mental and physical health of individuals by encouraging longer work hours, creating an unclear distinction between work and home, and reducing support from the organization. Furthermore, *Wang et al. (2021)* point out that working from home does not necessarily lead to greater balance between work and family. Therefore, it cannot be said that working from home is without disadvantages. Rather, the many factors that potentially contribute to effective work must be considered.

Findings elsewhere indicate that differences in the work environment may affect the quality of work among telecommuters. *Umishio et al. (2022)* report that telecommuters are frequently dissatisfied with the lighting, space, and IT environment, while being more satisfied with the temperature, air conditioning, and sound environment compared to office workers. *Natomi, Kato & Matsushita (2022)* report that when workers reside with housemates, noise problems occur regardless of the type of residence, and insufficient privacy can lead to higher stress. It has also been reported that the physical arrangement of the office and work environment, encompassing chairs, keyboards, and mice used for work, is a determinant of physical strain (*Rodringues et al., 2017*; *Celik et al., 2018*; *Mohammadipour et al., 2018*). There is concern that physical strain may also increase if a proper office area is not set up at home (*Garci et al., 2022*).

Based on the studies described above, it appears that the research results on the impact of telecommuting were not changed significantly by the pandemic (*Fílardí, de Castro & Zaníní, 2020*; *Kim et al., 2020*; *Song & Gao, 2020*). However, as highlighted by *Wang et al. (2021)*, telecommuting is, at times, undesirable for workers as it sometimes occurs when the home-based work environment is inadequate. The negative impact of telecommuting in such situations is a characteristics of post-pandemic telecommuting. *Antunes et al. (2023)* reviewed studies of teleworking, for both part-time and full-time workers, before and after the pandemic. The studies revealed that while most teleworkers were part-time before the pandemic, many were full-time during the pandemic, creating differences in psychosocial risk factors for teleworking. They suggested that part-time telework may positively impact work-family balance and social relationships. In contrast, the impacts may be different in cases of full-time telework.

Therefore, ergonomic awareness is essential in constructing an appropriate telecommuting environment. In Japan, the Ministry of Health, Labor, and Welfare, the Japan Human Factors and Ergonomics Society, and the Japan Society for Office Studies published guidelines on appropriate telecommuting environments (*Factors & Ergonomics Society, 2022*; *Ministry of Health, Labor, and Welfare of Japan, 2023*; *Office*

*Ergonomics Research Group, 2021*). However, as *Siqueria et al. (2023)* point out, many of the workers who transitioned to telecommuting during the pandemic did not possess the appropriate equipment infrastructure at home. Therefore, creating a work environment at home that meets the furniture, lighting, air conditioning, and all other requirements recommended in the guidelines may not be straightforward. Creating a more comfortable telecommuting environment will be made easier if the degree of importance of each factor is better understood.

When rating the importance of factors, it is necessary to take a multifaceted perspective and examine the impact of each factor on overall indicators, for example, comfort level. However, to the best of our knowledge, existing studies that examine the effectiveness of telecommuting rely on a single-factor approach. The ergonomic guidelines for telecommuting environments in Japan focus on physical facilities such as computers, other IT equipment, furniture, and lighting and how best to reduce physical burden. However, the author argues that telecommuting should be viewed as a system that can be analyzed using the SHEL model (*Hawkins, 1987*). This model incorporates elements that may be considered for living environment factors multidirectionally, including Software (work content, life rhythm, *etc.*), Hardware (furniture, equipment, *etc.*), Environment (indoor environment), and Liveware (relationships with housemates). By examining the factors that affect telecommuting satisfaction and the magnitude of their effects on various dependent variables, the author hopes to contribute to a better understanding of how to apply the guidelines.

Clearly, there is an incomplete understanding of how different aspects of the living environment, such as those captured by the SHEL model, influence the telecommuting experience. Therefore, the research question of this study is: "What life environment factors affect the effectiveness and comfort of telecommuting when the telecommuting environment is viewed from the perspective of the SHEL model?" A multiple linear regression model is used to analyze this research question. Specifically, the study examines satisfaction, stress, work engagement, and perceived productivity as indicators of the effectiveness and comfort of telecommuting. Factors in the living environment that may affect these indicators are comprehensively examined from the perspective of the SHEL model. This study is an in-depth examination of a preliminary report by the author (*Hawkins, 1987*) with an increased sample size and evaluation indices.

## MATERIALS & METHODS

The research methodology followed *Doi (2023)*, except for the target sample and the additional questions.

### Participants

A web-based survey was administered to 500 workers living in Japan. Participants were recruited *via* an Internet research company between 2023/4/21 and 2023/4/27. All participants were selected after working from home for at least two days a week for at least six months. Before answering the questionnaire, the consent to participate in the survey was obtained from the participants *via* web form. The participants were required to check
the box after they agree to answer the questionnaire. The survey was conducted following approval from the Ethics Committee of the Graduate School of Human Life and Ecology, Osaka Metropolitan University (Approval no. (23-10)).

## Questionnaire items

In this study, the effectiveness and comfort of the telecommuting environment were captured using several indicators. These included overall satisfaction with the telecommuting work style, satisfaction with the physical indoor environment, work engagement, perceived productivity, and stress levels. Each item was analyzed as a dependent variable in the model. The following questions were used to measure each indicator. This study added items on perceived productivity to the questionnaire of *Doi (2023)* and kept the other items the same.

### Satisfaction

The participants were asked to indicate their overall satisfaction with their telecommuting work style and satisfaction with the physical environment in which they work. Responses were captured using a five-point Likert scale.

### Work engagement

Work engagement is a positive and fulfilling psychological state. The Utrecht Work Engagement Scale is proposed by *Schaufeli & Bakker (2010)* as a measure of work engagement. In this study, the Japanese version of the Utrecht Work Engagement Scale (simplified version) was used (*Shimazu et al., 2008*; *Schaufeli et al., 2019*). This scale elicited responses to three questions about vitality, enthusiasm, and immersion using a seven-point rating scale. The total score across the three items was recorded to form an overall score.

### Stress reactions

In Japan, a Brief Job Stress Questionnaire was used to assess stress at work (*Shimomitsu et al., 2000*; *Shimomitsu, 2000*; *Shimomitsu, 1998*; *Ministry of Health & Welfare of Japan, 2023*). This questionnaire consisted of three parts: job stressors, stress reactions, and modifiers, and was used to identify high-stress individuals. In this study, 29 items related to stress were used. These 29 items consisted of a four-point rating scale based on liveness, irritability, fatigue, anxiety, depression, and somatic complaints. The total score (after reversing the negative items) was recorded and formed an overall stress score.

### Perceived Productivity

The following three questions were posed. Participants responded using a five-point scale based on their work situation over the previous month:

- Do you think your work productivity is high?
- Do you think you are achieving more than expected?
- Do you think telecommuting has improved your work efficiency?

The total score of these three items was recorded and formed the perceived productivity score.

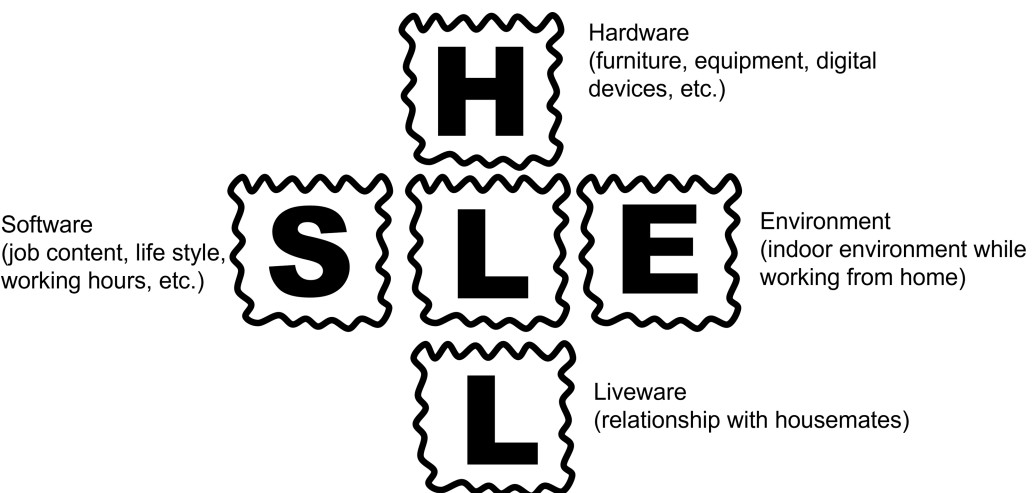

**Figure 1** Framework of the SHEL model (modified from *Hawkins, 1987*).

The SHEL model was originally proposed by *Hawkins (1987)* as a model to understand human error among aircrew members. The model is depicted in Fig. 1. It explains that errors occur due to incompatibility between the central part (Liveware) and the surrounding S (Software), H (Hardware), E (Environment), and L (Liveware) (gaps between the central L and other surrounding factors) factors. This model was originally proposed as a means of analyzing human error analysis, but it may also be used to capture the suitability of human-based systems. In this study, the author considers the compatibility between the telecommuting worker as a central L and the surrounding S, H, E, and L based on the SHEL model. Several explanatory variables are included to investigate the telecommuting environment. The questions are described below.

### Software

Intangible items related to working and living when telecommuting were defined as software. These mainly included questions on work content, life rhythm, and working hours. Specifically, participants were asked about a range of factors, including job autonomy, the number of telecommuting days per week, the frequency and duration of online meetings, daily work hours and breaks, the qualitative workload, the quantitative workload, whether regular breaks were taken, whether participants ate at regular times, the frequency of interruptions, the time spent on household tasks, and the frequency of overtime work.

A total score for "job autonomy" was calculated by measuring the following three items, drawn from a study by Morinaga (*Takahashi & Kato, 2022*): "I can work at my own pace", "I can decide the order and method of work by myself", and "I need to communicate with other people (inversion). For "qualitative workload" and "quantitative workload", the corresponding items (three items each) from the Brief Job Stress Questionnaire (*Shimomitsu et al., 2000*) were used. For questions about regular breaks, whether participants ate at regular times, and whether their working hours were constant, a five-point Likert scale

**Table 1  Five-point scales for the items related to working hours and meeting time.**

| Questions | 1 | 2 | 3 | 4 | 5 |
|---|---|---|---|---|---|
| Number of telecommuting days per week | Less than 2 days | 3 days | 4 days | 5 days | More than 6 days |
| Number of online meetings per week | 0 to 1 time | 2 to 4 times | 5 to 9 times | 10 to 14 times | More than 15 times |
| Online meeting hours per day | Less than 30 min | More than 30 min, less than 2 h | More than 2 h, less than 4 h | More than 4 h, less than 6 h | More than 6 h |
| Telecommuting hours per day | Less than 4 h | More than 4 h, less than 6 h | More than 6 h, less than 8 h | More than 8 h, less than 10 h | More than 10 h |
| Breaks per day | Less than 30 min | More than 30 min, less than 1 h | More than 1 h, less than 1.5 h | More than 1.5 h, less than 2 h | More than 2 h |

(1: not at all applicable, 5: very applicable) was used. Frequency-related questions, such as the interruption of work for household tasks and overtime work, were measured using a five-point Likert scale ranging from 1 (indicating rarely) to 5 (indicating frequently). The items related to working hours and meeting time were measured using a five-point scale, as shown in Table 1.

### Hardware

Questions on the use of furniture and equipment by telecommuting are listed. Specifically, participants were asked about the size of their desk, whether they had a personal chair (not shared chair), whether the chair had armrests, whether the chair had an adjustment function, perceived Internet speed, perceived PC performance, whether they owned a PC monitor, the size of the PC monitor, and whether they had external speakers, a microphone, or a headset.

Categorical data (yes = 1, no = 0) was collected for the binary questions, while the remainder were quantified using a five-point Likert scale. Perceived Internet speed and PC performance were not measured by physical values but by subjective ratings, such as "1: very slow, 2: slow, 3: undecided, 4: fast, 5: very fast",. The PC monitor sizes were categorized as follows: "1: less than 13 inches, 2: about 14∼17 inches, 3: about 18∼22 inches, 4: about 23∼27 inches, and 5: 28 inches or larger.

### Environment

Questions about the physical environment of the indoor space were also included. Participants were asked about the perceived brightness of the room, the subjective level of ambient noise, the subjective size of the workroom, whether they had a dedicated workspace, whether their workspace was within the space used for housework and daily life, and whether they had a space for eating and resting outside the workspace.

The participants were asked to rate the "brightness of the room" and "ambient noise" on a five-point scale from 1: dark to 5: bright enough. Questions regarding the presence or absence of other items were quantified as categorical data (yes = 1, no = 0).

*Liveware*

Questions about the relationship with housemates were listed. The participants were asked about the presence and attributes of their housemates, whether their housemates were in the same room when they were working from home, the frequency of interruption by housemates while at work, and the degree to which they were responsible for their household tasks.

The participants were asked whether they lived with their partner, preschooler, elementary school student, child of junior high school age or older, child who had completed schooling, or parent. The participants were asked to rate on a five-point scale from 1: rarely to 5: frequently, "whether the housemates are in the same room during work" and "the degree of intervention from the roommate", and to rate the "degree of sharing household tasks with the housemate" on a five-point scale from 1: almost no sharing (the roommate does it), 2: doing it him/herself but the housemate does it more, and 3: doing about the same amount of housework as my housemate, 4: doing a large amount of housework myself, and 5: doing almost all of the housework myself.

## Data analysis

Multiple linear regression analysis was used to analyze the relationship between the effectiveness of telecommuting and life environment factors. Five regression models were specified for each objective variable, using five indicators of overall effectiveness (overall satisfaction, satisfaction with the physical working environment, work engagement, stress, and perceived productivity). Several other control variables (age, gender, and job) were also included. In all cases, explanatory variables were selected using a forward-backward stepwise method. The criterion for variable selection was a $p$-value equal to 0.2 for both inputs and removals. In the results tables, $b$ refers to the partial regression coefficient, and $\beta$ refers to standard partial regression analysis.

## RESULTS

This study included 500 workers (mean 43.85 years, SD: 110.71, 250 males and 250 females) living in Japan. The questionnaire includes items related to the use of chairs and desks when working at home, as well as questions about housemates. To examine the effects of these factors, respondents answering that they did not use chairs or desks and had no housemates were excluded. The samples included in this analysis totaled 336 (mean 43.26 years, SD = 10.51, 166 males and 170 females) workers who use a chair/desk when working at home and who reside with a housemate. There was no missing data for these 336 samples. A comparison of the 164 removed samples and the 336 samples used in the study for differences in demographics showed no significant differences in age ($t$ (498) = 1.805, $p = 0.078$), gender ($z = 0.381$, $p = 0.703$), or the proportion of managers ($z = -1.592$, $p = 0.111$).

Table 2 summarizes the results of the responses to each question, showing the mean and standard deviation for the questions that relied on a five-point rating scale and the frequencies of 1 and 0 for items coded with categorical data.

**Table 2  Summary of the questionnaire results.**

| Questions | Mean | SD | Questions | n | |
|---|---|---|---|---|---|
| | | | | **1 (yes)** | **0 (no)** |
| Age | 43.25 | 10.51 | Gender (1=Male) | 166 | 170 |
| Overall satisfaction | 3.72 | 1.25 | Job (0=Managerial position) | 220 | 116 |
| Satisfaction with the physical environment | 3.46 | 1.17 | Dedicated desk | 233 | 103 |
| Perceived productivity | 9.5 | 4.1 | Dedicated chair | 240 | 96 |
| Work engagement | 11.26 | 4.95 | Chair: armrests | 189 | 147 |
| Stress reactions | 51.7 | 22.04 | Chair: adjustment | 207 | 129 |
| Job autonomy | 10.2 | 4.03 | External monitor | 156 | 180 |
| Qualitative workload | 6.73 | 2.5 | External speaker | 106 | 230 |
| Quantitative workload | 6.66 | 2.38 | External microphone | 127 | 209 |
| Brightness of workroom | 3.51 | 1.07 | Headset | 151 | 185 |
| Ambient noise | 3.24 | 1.1 | Dedicated workspace | 201 | 135 |
| Size of workroom | 3.09 | 1.02 | Workspace is within the space they use for housework and daily life | 245 | 91 |
| Size of desk | 3.01 | 1.12 | Space for eating and resting | 278 | 58 |
| Internet speed | 3.18 | 0.99 | Living with partner | 229 | 107 |
| PC performance | 3.27 | 1.08 | Living with preschooler | 68 | 268 |
| Size of PC monitor | 2.54 | 1.05 | Living with elementary school students | 89 | 247 |
| Frequency of being in the same room with housemates | 2.07 | 1.23 | Living with junior high school students or older | 79 | 257 |
| Frequency of intervention by housemates | 2.21 | 1.13 | Living with children who have completed their studies | 33 | 303 |
| Degree of housework sharing with housemates | 2.82 | 1.27 | Living with parents | 88 | 248 |
| Regularly breaks | 3.56 | 1.06 | | | |
| Regularly meals | 3.66 | 1.1 | | | |
| Frequency of interrupted work for household tasks | 2.59 | 1.05 | | | |
| Time spent on household tasks | 2.28 | 1.1 | | | |
| Regularly worktime | 3.39 | 1.21 | | | |
| Frequency of thinking about work after work hours | 2.84 | 1.09 | | | |
| Frequency of overtime working | 2.69 | 1.15 | | | |
| Frequency of looking at a PC for work on days off | 2.55 | 1.21 | | | |
| Number of telecommuting days per week | 2.64 | 1.27 | | | |
| Number of online meetings per week | 2.23 | 1.09 | | | |
| Online meeting hours per day | 2 | 0.83 | | | |
| Telecommuting hours per day | 3.07 | 0.92 | | | |
| Breaks per day | 2.62 | 0.87 | | | |

| Table 3 Correlation coefficients among objective variables. | | | | | |
|---|---|---|---|---|---|
| | **Overall satisfaction** | **Satisfaction with the physical environment** | **Perceived productivity** | **Work engagement** | **Stress reactions** |
| Overall satisfaction | – | 0.7 | 0.31 | 0.08 | −0.21 |
| Satisfaction with the physical environment | ** | – | 0.4 | 0.24 | −0.21 |
| Perceived productivity | ** | ** | – | 0.32 | −0.16 |
| Work engagement | | ** | ** | – | −0.05 |
| Stress reactions | ** | ** | ** | | – |

**Notes.**
$^{**}p < 0.01$.
$^{*}p < 0.05$.

Correlation coefficients were then calculated to confirm the relationships among the objective variables (Table 3). A strong positive correlation between overall satisfaction with telecommuting and the indoor environment was identified. Perceived productivity was positively correlated with overall satisfaction, satisfaction with the indoor environment, and work engagement. In contrast, the stress levels showed almost no correlation with any of the other items.

The following section presents the results of multiple regression analysis for each objective variable. Table 4 shows the results of the multiple regression analysis using overall satisfaction with the telecommuting work style as the independent variable. The adjusted-$R^2$ value for this model was 0.31 ($F_{(17,318)} = 9.81$, $p < 0.01$). The VIF of each selected explanatory variable was at most 2.02, indicating no multicollinearity problem. In this model, the standard partial regression coefficient of job autonomy ($\beta = 0.31$, confidence interval of $b$: 0.1~0.19) was particularly high and most significantly affected overall satisfaction.

Table 5 shows the results of a multiple regression analysis using satisfaction with the telecommuting indoor environment as the objective variable. The adjusted $R^2$ value for this model was 0.31 ($F_{(12,323)} = 13.33$, $p < 0.01$). The VIFs for each selected explanatory variable were all less than 2, indicating no multicollinearity problem. In this model, the standard partial regression coefficients for brightness of workroom ($\beta = 0.25$, confidence interval of $b$: 0.16~0.39) and job autonomy ($\beta = 0.2$, confidence interval of $b$: 0.04~0.14) were particularly high and had the largest influence on satisfaction with the telecommuting indoor environment.

Table 6 shows the results of the multiple regression analysis using work engagement as the independent variable. The adjusted $R^2$ value for this model was 0.37 ($F_{(19,316)} = 11.30$, $p < 0.01$). The VIF of each selected explanatory variable was at most 2.02, indicating no multicollinearity problem. In this model, the standard partial regression coefficient of job autonomy ($\beta = 0.34$, confidence interval of $b$: 0.38~0.69) was notably high and had the largest effect on work engagement.

Table 7 shows the results of the multiple regression analysis in which perceived productivity formed the independent variable. The adjusted $R^2$ value for this model

**Table 4  Result of multiple regression analysis: overall satisfaction.**

| Predictors | $b$ | $\beta$ | 95% confidence interval of $b$ | | $t$ | | VIF |
|---|---|---|---|---|---|---|---|
| | | | Lower | Upper | | | |
| Age | −0.01 | −0.12 | −0.03 | 0 | −2.22 | * | 1.39 |
| Gender | 0.21 | 0.09 | −0.04 | 0.46 | 1.66 | | 1.27 |
| Job autonomy | 0.15 | 0.31 | 0.1 | 0.19 | 5.86 | ** | 1.31 |
| Qualitative workload | 0.04 | 0.07 | −0.01 | 0.1 | 1.46 | | 1.19 |
| Brightness of workroom | 0.19 | 0.16 | 0.07 | 0.3 | 3.13 | ** | 1.26 |
| Size of PC monitor | 0.15 | 0.13 | 0.04 | 0.27 | 2.56 | * | 1.19 |
| External speaker | −0.45 | −0.17 | −0.79 | −0.1 | −2.57 | * | 2.02 |
| External microphone | 0.39 | 0.15 | 0.06 | 0.71 | 2.32 | * | 2.02 |
| Living with partner | 0.26 | 0.1 | −0.01 | 0.53 | 1.86 | | 1.3 |
| Living with preschooler | −0.37 | −0.12 | −0.7 | −0.04 | −2.23 | * | 1.38 |
| Living with junior high school students or older | −0.29 | −0.1 | −0.57 | 0 | −2 | * | 1.14 |
| Frequency of being in the same room with housemates | −0.1 | −0.1 | −0.2 | 0 | −1.93 | | 1.21 |
| Time spent on household tasks | −0.14 | −0.12 | −0.26 | −0.02 | −2.28 | * | 1.36 |
| Frequency of overtime working | 0.16 | 0.15 | 0.03 | 0.29 | 2.34 | * | 1.87 |
| Frequency of looking at a PC for work on days off | −0.17 | −0.16 | −0.3 | −0.04 | −2.57 | * | 1.98 |
| Number of telecommuting days per week | 0.08 | 0.08 | −0.02 | 0.17 | 1.61 | | 1.19 |
| Online meeting hours per day | −0.11 | −0.07 | −0.26 | 0.04 | −1.39 | | 1.25 |
| Constant | 1.76 | | 0.78 | 2.74 | 3.54 | ** | |

Notes.
** $p < 0.01$.
* $p < 0.05$.
$b$: partial regression coefficient, $\beta$: standard partial regression coefficient.

was 0.40 ($F$ (16,319) = 15.11, $p < 0.01$). The VIF of each selected explanatory variable was at most 2.12, indicating no multicollinearity problem. In this model, the standard partial regression coefficient of job autonomy ($\beta = 0.35$, confidence interval of $b$: 0.26∼0.47) was notably high and had the largest effect on perceived productivity.

Table 8 shows the results of the multiple regression analysis in which stress level formed the independent variable. The adjusted R2 for this model was 0.37 ($F$ (16,319) = 13.31, $p < 0.01$). The VIFs for each selected explanatory variable were all less than 2, indicating no multicollinearity problem. In this model, the standard partial regression coefficients of frequency of interrupted work for household tasks ($\beta = 0.26$, confidence interval of $b$: 2.69∼3.24) and gender ($\beta = -0.2$, confidence interval of $b$: -10.1∼−3.78) were notably high and significantly affected stress level.

## DISCUSSION

In this study, a total of 164 study participants who do not use a desk/chair when working from home and live without housemates were excluded from the analysis. Compared to the 336 participants used in the analysis, several demographic factors (age, gender, and manager) showed no significant differences to the excluded group. This suggests that the sample is not biased toward any particular characteristic. However, it is likely that the
**Table 5  Result of multiple regression analysis: satisfaction with the physical indoor environment.**

| Predictors | b | β | 95% confidence interval of b | | t | | VIF |
|---|---|---|---|---|---|---|---|
| | | | Lower | Upper | | | |
| Job autonomy | 0.09 | 0.2 | 0.04 | 0.14 | 3.63 | ** | 1.48 |
| Qualitative workload | 0.04 | 0.07 | −0.02 | 0.09 | 1.39 | | 1.09 |
| Brightness of workroom | 0.28 | 0.25 | 0.16 | 0.39 | 4.66 | ** | 1.42 |
| PC performance | −0.1 | −0.09 | −0.21 | 0.02 | −1.64 | | 1.45 |
| Size of PC monitor | 0.1 | 0.09 | −0.01 | 0.2 | 1.77 | | 1.14 |
| Dedicated workspace | 0.41 | 0.17 | 0.19 | 0.62 | 3.64 | ** | 1.05 |
| Living with preschooler | −0.2 | −0.07 | −0.48 | 0.08 | −1.4 | | 1.16 |
| Living with junior high school students or older | −0.28 | −0.1 | −0.53 | −0.03 | −2.16 | * | 1.05 |
| Regularly meals | 0.18 | 0.17 | 0.06 | 0.3 | 2.92 | ** | 1.64 |
| Frequency of interrupted work for household tasks | −0.09 | −0.08 | −0.19 | 0.02 | −1.66 | | 1.08 |
| Online meeting hours per day | −0.17 | −0.12 | −0.31 | −0.03 | −2.42 | * | 1.21 |
| Telecommuting hours per day | 0.12 | 0.1 | 0 | 0.25 | 1.93 | | 1.19 |
| Constant | 0.71 | | −0.1 | 1.52 | 1.72 | | |

**Notes.**
** $p < 0.01$.
* $p < 0.05$.
b: partial regression coefficient, β: standard partial regression coefficient.

types of telecommuting jobs with no requirement of the use of a desk or chair are different from those analyzed in this experiment. If this is the case, the findings of this study may not be applicable to these types of jobs. The same findings are obtained for the presence or absence of a housemate. In this study, participants without a housemate were omitted in order to focus on the role and impact housemates. The presence of a housemate may impacting the telecommuting experience, and therefore, future research is required.

The coefficients of determination for each of the multiple regression models calculated in this study were approximately 0.3~0.4, indicating that the models do not have high explanatory power. However, the significance of the regression equations was confirmed, suggesting that appropriate models were obtained when examining factors affecting each independent variable.

First, the model pertaining to overall satisfaction is described. The standard partial regression coefficient of job autonomy was found to be remarkably high for overall satisfaction. In other words, overall satisfaction in telecommuting is more influenced by the software aspect of job content than by the physical environment of the room. Other explanatory variables that were found to be significant included brightness of lighting, monitor size, external microphone, and frequency of overtime, all of which were positively correlated. Meanwhile, external speakers, living with a preschooler, living with a child of junior high school age or older, time spent on household tasks, and frequency of looking at a computer for work on weekends and holidays were all negatively correlated to overall satisfaction. The result shows that the greater the frequency of overtime, the greater the overall satisfaction. This suggests that among those who exhibit higher overall satisfaction, more work is done outside of working hours. In addition, the author identifies several

**Table 6 Result of multiple regression analysis: work engagement.**

| Predictors | b | β | 95% confidence interval of b | | t | | VIF |
|---|---|---|---|---|---|---|---|
| | | | Lower | Upper | | | |
| Age | 0.04 | 0.12 | 0.01 | 0.08 | 2.34 | * | 1.29 |
| Gender | −0.85 | −0.1 | −1.66 | −0.05 | −2.08 | * | 1.34 |
| Job autonomy | 0.53 | 0.34 | 0.38 | 0.69 | 6.7 | ** | 1.38 |
| Qualitative workload | −0.36 | −0.19 | −0.54 | −0.18 | −3.97 | ** | 1.18 |
| Brightness of workroom | 0.43 | 0.11 | 0.03 | 0.83 | 2.12 | * | 1.5 |
| Ambient noise | −0.31 | −0.08 | −0.68 | 0.05 | −1.69 | | 1.32 |
| Dedicated desk | −0.61 | −0.07 | −1.5 | 0.28 | −1.35 | | 1.39 |
| Size of desk | 0.31 | 0.09 | −0.07 | 0.69 | 1.62 | | 1.49 |
| Size of PC monitor | 0.44 | 0.11 | 0.08 | 0.8 | 2.39 | * | 1.16 |
| Headset | 0.62 | 0.08 | −0.14 | 1.37 | 1.6 | | 1.17 |
| Dedicated workspace | 1.4 | 0.17 | 0.57 | 2.23 | 3.33 | ** | 1.35 |
| Living with partner | −0.69 | −0.08 | −1.51 | 0.13 | −1.66 | | 1.21 |
| Frequency of intervention by housemates | 0.36 | 0.1 | −0.03 | 0.76 | 1.81 | | 1.65 |
| Frequency of interrupted work for household tasks | −0.52 | −0.14 | −0.93 | −0.12 | −2.55 | * | 1.49 |
| Frequency of overtime working | 0.55 | 0.16 | 0.12 | 0.98 | 2.53 | * | 2.02 |
| Frequency of looking at a PC for work on days off | 0.41 | 0.12 | 0.01 | 0.81 | 2.04 | * | 1.92 |
| Number of online meetings per week | −0.35 | −0.09 | −0.76 | 0.05 | −1.73 | | 1.58 |
| Online meeting hours per day | 0.92 | 0.19 | 0.4 | 1.43 | 3.47 | ** | 1.52 |
| Telecommuting hours per day | −0.42 | −0.1 | −0.85 | 0.01 | −1.94 | | 1.28 |
| Constant | 2.68 | | −0.62 | 5.98 | 1.6 | | |

**Notes.**
** $p < 0.01$.
* $p < 0.05$.
b: partial regression coefficient, β: standard partial regression coefficient.

factors that potentially reduce overall satisfaction that are related to the worker's own family situation, such as the status of cohabitation and time spent on household tasks.

As for satisfaction with the physical indoor environment, the standard partial regression coefficients for the brightness of lighting, job autonomy, whether the worker has a dedicated workspace, and whether the worker eats meals at regular times were large. The results suggest that it is important to have sufficient lighting and a workspace. On the other hand, the results also revealed the importance of other factors, such as job autonomy and the ability to eat meals regularly, for satisfaction. In addition, it was also shown that satisfaction with the indoor environment tends to decrease with the amount of time spent in online meetings.

As for work engagement, the standard partial regression coefficients indicated that job autonomy, low qualitative workload, a dedicated workspace, and the length of online meeting time per day are significant factors. As mentioned above, the length of online meeting time was found to be correlated with satisfaction with the indoor environment. However, the results suggest that more online meeting time is associated with higher work engagement. Telecommuting employees may have fewer opportunities to talk with

**Table 7 Result of multiple regression analysis: perceived productivity.**

| Predictors | b | β | 95% confidence interval of b | | t | | VIF |
|---|---|---|---|---|---|---|---|
| | | | Lower | Upper | | | |
| Job autonomy | 0.37 | 0.35 | 0.26 | 0.47 | 6.94 | ** | 1.4 |
| Qualitative workload | −0.21 | −0.16 | −0.32 | −0.09 | −3.57 | ** | 1.14 |
| Brightness of workroom | 0.38 | 0.15 | 0.12 | 0.63 | 2.89 | ** | 1.42 |
| Size of desk | 0.39 | 0.16 | 0.16 | 0.63 | 3.24 | ** | 1.38 |
| Size of PC monitor | 0.36 | 0.14 | 0.13 | 0.6 | 3.06 | ** | 1.15 |
| Headset | 0.38 | 0.07 | −0.11 | 0.87 | 1.54 | | 1.13 |
| Dedicated workspace | 0.48 | 0.09 | −0.01 | 0.97 | 1.94 | | 1.11 |
| Workspace is within the space they use for housework and daily life | 0.49 | 0.08 | −0.04 | 1.02 | 1.82 | | 1.05 |
| Living with preschooler | −0.65 | −0.1 | −1.27 | −0.04 | −2.1 | * | 1.16 |
| Living with junior high school students or older | −0.39 | −0.06 | −0.95 | 0.17 | −1.36 | | 1.08 |
| Living with parents | −0.48 | −0.08 | −1.03 | 0.07 | −1.73 | | 1.11 |
| Frequency of intervention by housemates | 0.26 | 0.11 | 0 | 0.51 | 1.95 | | 1.63 |
| Frequency of interrupted work for household tasks | −0.38 | −0.15 | −0.65 | −0.12 | −2.85 | ** | 1.49 |
| Regularly worktime | 0.24 | 0.1 | 0.03 | 0.44 | 2.28 | * | 1.18 |
| Frequency of thinking about work after work hours | −0.22 | −0.09 | −0.51 | 0.08 | −1.45 | | 1.98 |
| Frequency of overtime working | 0.32 | 0.13 | 0.03 | 0.61 | 2.16 | * | 2.11 |
| Constant | 3.4 | | 1.68 | 5.11 | 3.9 | ** | |

**Notes.**
$^{**}p < 0.01$.
$^{*}p < 0.05$.
b: partial regression coefficient, β: standard partial regression coefficient.

colleagues (*Marshall, Michaels & Mulki, 2007*), and it is thought that communicating with coworkers through online conferencing positively affects work engagement (*Takahashi & Kato, 2022*). In addition, a positive correlation was also found between the frequency of overtime work and the frequency of looking at a PC on weekends and holidays. It is natural to assume that there is an inverse causal relationship, *i.e.,* that high work engagement allows workers to tolerate working outside of their working hours. The results also suggest a negative correlation between the frequency of interruptions due to housework and work engagement, indicating that work engagement is affected by the content of work and the family situation.

As for perceived productivity, the standard partial regression coefficients for each variable indicated that job autonomy, low qualitative workload, brightness of lighting, desk size, monitor size, and low frequency of interruptions due to household tasks are significant explanatory factors. In terms of productivity, factors related to the physical fulfillment of the living space, such as lighting, desk size, and monitor size, are important. Similarly, job autonomy, qualitative workload, and frequency of interruptions due to housework are also found to be important in terms of whether the environment is conducive to concentrating on work.

**Table 8 Result of multiple regression analysis: stress reactions.**

| Predictors | b | β | 95% confidence interval of b | | t | | VIF |
|---|---|---|---|---|---|---|---|
| | | | Lower | Upper | | | |
| Gender | −6.94 | −0.2 | −10.1 | −3.78 | −4.32 | ** | 1.14 |
| Job autonomy | −0.49 | −0.07 | −1.19 | 0.21 | −1.38 | | 1.54 |
| Brightness of workroom | −1.39 | −0.09 | −3.17 | 0.4 | −1.53 | | 1.65 |
| Ambient noise | −2.99 | −0.19 | −4.55 | −1.42 | −3.76 | ** | 1.36 |
| Size of desk | −1.16 | −0.08 | −2.75 | 0.43 | −1.44 | | 1.47 |
| Dedicated chair | −5.12 | −0.13 | −9.15 | −1.09 | −2.5 | * | 1.52 |
| Chair: armrests | 2.74 | 0.08 | −1.15 | 6.63 | 1.38 | | 1.71 |
| Chair: adjustment | 5.18 | 0.15 | 1.16 | 9.2 | 2.54 | * | 1.75 |
| Internet speed | −2.34 | −0.13 | −4.18 | −0.49 | −2.49 | * | 1.52 |
| Living with partner | −2.52 | −0.07 | −5.87 | 0.83 | −1.48 | | 1.12 |
| Frequency of being in the same room with housemates | 1.27 | 0.09 | −0.05 | 2.59 | 1.89 | | 1.22 |
| Regularly meals | 1.66 | 0.11 | −0.12 | 3.45 | 1.83 | | 1.78 |
| Frequency of interrupted work for household tasks | 4.23 | 0.26 | 2.69 | 5.78 | 5.39 | ** | 1.22 |
| Frequency of thinking about work after work hours | 1.75 | 0.11 | 0.26 | 3.24 | 2.31 | * | 1.22 |
| Online meeting hours per day | 2.57 | 0.12 | 0.55 | 4.59 | 2.5 | * | 1.28 |
| Breaks per day | 1.18 | 0.06 | −0.62 | 2.98 | 1.29 | | 1.13 |
| Constant | 59.25 | | 49.15 | 69.35 | 11.54 | ** | |

**Notes.**
** $p < 0.01$.
* $p < 0.05$.
b: partial regression coefficient, β: standard partial regression coefficient.

The results reveal the importance of job autonomy. This is in line with *Takahashi & Kato (2022)* who find that employees' active job creation and supervisor support are more important than introducing a mobile environment. In the design of telecommuting jobs, the impact of job content is more significant than that of hardware or home conditions, and jobs with a high degree of job autonomy are relatively suited to telecommuting. In addition, the brightness of lighting and the presence or absence of a workspace dedicated to one's use are also considered to have a relatively high impact on many indicators, suggesting that improvements to the living environment are key. The report on the importance of personal space in the telecommuting environment is consistent with the findings of *Tleuken et al. (2022)*.

Next, the author turns to the analysis of stress levels. The standard partial regression coefficients for each variable indicate that the effects of gender (males have a lower stress reaction), environmental noise, and frequency of interruptions due to household chores are significant determinants of stress. *Gimenz-Nadal, Molina & Velilla (2020)* also note the existence of gender differences, reporting that male teleworkers showed significantly lower stress when working at home, while no significant differences were observed for females. The same relationship is evident in this study. *Wang et al. (2021)* also found that noise had a significant impact on stress, which is consistent with our findings. The standard partial regression coefficients for the frequency of interruptions due to housework were

relatively high, suggesting a significant relationship with stress. In Japan, women more often perform housework compared to men (*Gender Equality Bureau Cabinet Office of Japan, 2020*), which may explain the difference in stress between genders. Whether or not the worker possessed a chair and whether the chair had an adjustment function was found to be significantly related to stress. The results highlight the importance of physical comfort on the well-being of workers.

Given the results, it is evidently important to consider job autonomy in relation to telecommuting environments. The most important factor of workspace facilities appears to be the provision of a dedicated workspace with sufficient consideration of lighting, environmental noise, and chairs. In response to the research question in this study, when considering the living environment for telecommuting from the perspective of the SHEL model, these factors, in particular, are considered to have an important influence on the ease of telecommuting.

Finally, the author turns to the limitations of the study. The most important limitation is the limited sample size. This study did not analyze the potential impact of cultural differences nor the number of days worked at home. Results may differ between workers who work at home part-time and those who work at home full-time. In addition, because this study emphasizes a multifaceted approach *via* the SHEL model, each element of the model must be examined in detail. For example, *Garci et al. (2022)* examine items that impact subjective physical discomfort among telecommuters and find that a laptop stand affects head and eye comfort. In addition, sitting for more than eight hours affects hand comfort, and an uncomfortable desk affects back and neck comfort. Therefore, the comfort of different parts of the body should be considered individually. To utilize the findings of this study to inform future ergonomics guidelines, it is necessary to integrate the findings of this study with additional research.

## CONCLUSIONS

This study investigates how the at-home work environment affects satisfaction, work engagement, perceived productivity, and stress. The author examines a range of factors categorized as software (work content, life rhythm, *etc*.), hardware (furniture, equipment, *etc*.), environment (indoor environment), and liveware (relationships with housemates). Using the multifactorial approach of the SHEL model, this study examined how different elements in the living environment of a telecommuter affect the effectiveness and comfort of work. Multiple linear regression analysis is performed to analyze the effects of each explanatory variable from the viewpoint of the SHEL model. The results suggest that a high degree of job autonomy and the availability of a dedicated workspace for a worker yields a significantly positive impact on satisfaction and work engagement. Negative issues, namely stress, are significantly impacted by factors including environmental noise, interruptions due to housework, and the availability of an ergonomic chair.

### Funding

This research was funded by JSPS KAKENHI, grant number 22K18140. The funders had no role in study design, data collection and analysis, decision to publish, or preparation of the manuscript.

### Grant Disclosures

The following grant information was disclosed by the author:
JSPS KAKENHI: 22K18140.

### Competing Interests

The author declares that they have no competing interests.

### Author Contributions

- Toshihisa Doi conceived and designed the experiments, performed the experiments, analyzed the data, prepared figures and/or tables, authored or reviewed drafts of the article, and approved the final draft.

### Human Ethics

The following information was supplied relating to ethical approvals (*i.e.,* approving body and any reference numbers):

The Ethics Committee of the Graduate School of Human Life and Ecology, Osaka Metropolitan University (Approval no. 23-10).

### Data Availability

The raw data are available in the Supplementary File.

### Supplemental Information

Supplemental information for this article can be found online at http://dx.doi.org/10.7717/peerj.17301#supplemental-information.

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
