# Peer review of "The relationship between the living environment and remote working: an analysis using the SHEL model"

_PeerJ, doi:10.7717/peerj.17301_

## Round 0.1 · original submission · Major Revisions

As you can see, the reviewers have offered constructive feedback that I believe will greatly assist you in revising your manuscript. I kindly request that you provide comprehensive responses to each comment from the reviewers.

Reviewer 1 ·

Basic reporting

- Please show sample size in each of the tables
- Please add a footnote to describe what the b and the beta columns stand for in each of the regression tables. Please also indicate what regression was used (e.g., linear, logistic) in the tables' titles.
- Parts of the descriptions in the Participants section should go to the results section, including the number of participants, the number of samples included in the final model, and the demographics of the samples.

Experimental design

- Please indicate the duration of the participant recruitment period, and the inclusion and exclusion criteria. Why were only 336 out of 500 workers selected for the final study? To examine the potential of selection bias, could you compare the demographics and the objective variables of the 336 samples included in the study and the 164 samples excluded? A significant difference could indicate selection bias and should be mentioned in the discussion.
- Please describe how you dealt with missing data. Were there any missing data for the variables listed in Table 2? If yes, please indicate the number of samples for the continuous variables. Were there any missing data for objective variables? Please indicate the number of samples in each of the regression tables.
- Please elaborate on the findings of each regression table in the result section. After describing R2 and VIFs, you could add one or two sentences to describe what covariates are most significantly associated with the outcome variables and also include the corresponding regression coefficients and confidence intervals.
- In the method section, please indicate what regression was used in your study - Linear regression or logistic regression? Please also add one or two sentences to describe what the b and the beta columns stand for respectively in the regression tables.

Validity of the findings

no comment

·

Basic reporting

Dear author,

I appreciate the comprehensive exploration in this study, which delves into the correlation between living environment factors and satisfaction, work engagement, perceived productivity, and stress among teleworkers. However, I have a few suggestions to further improve this paper:

1.1 Clarity and Professional English Usage: The article is written in clear and professional English, adhering to the required technical language standards. The communication is effective and free from ambiguities.

1.2 Literature References and Field Context: The introduction provides an overview of the relationship between the remote work environment and factors such as satisfaction, work engagement, perceived productivity, and stress. While the article references several studies, it is noted that the introduction could differentiate the effects before, during, and after the pandemic regarding remote work. The author references studies conducted in different contexts, and it is crucial to address whether the results are consistent post-pandemic. When mentioning ergonomic guidelines, emphasizing the organization of work and its impact on the discussed objectives would enhance the article.

Experimental design

2.1 Research Question Definition: The research question is not explicitly stated in the introduction, and it is crucial to include it to provide clarity for readers. Additionally, the submission should clearly identify the knowledge gap being investigated. While the criteria "It is stated how research fills an identified knowledge gap" are met, a more explicit presentation of the research question and its relevance would strengthen the article. Provide statements on how the study contributes to filling the identified gap.

2.2 Coherence in Objectives: There is a discrepancy between the objective outlined in the abstract and the background. While the abstract states that the study investigates the relationship between living environment factors and teleworkers' satisfaction, work engagement, productivity, and stress, the background mentions the aim of creating an optimum environment for telecommuting workers. Creating an optimum environment involves additional variables like work organization, relationships with supervisors, the nature of work, and distinctions between partial and total telecommuting, which are not adequately addressed in this study. It is recommended to align the abstract and background to maintain coherence in the study's objectives.

The article could benefit from explicitly stating the research question, clarifying the knowledge gap, and ensuring consistency between the abstract and background regarding the study's objectives. This will enhance the overall coherence and effectiveness of the research presentation.

Validity of the findings

3.1 Impact and Novelty Assessment: The article appropriately acknowledges that impact and novelty are not assessed, and it encourages meaningful replication with a clear rationale and benefit to the literature. The study presents statistically significant findings, particularly regarding the organization of work, which is highlighted in the statement: "The results reveal the importance of job autonomy." This adds value to the literature and aligns with the journal's criteria for encouraging studies with valuable contributions. The study's emphasis on job autonomy aligns with existing research, reinforcing the significance of job content in telecommuting.

3.2 Data Availability and Robustness: All underlying data have been provided, demonstrating robustness, statistical soundness, and control. This meets the journal's requirement for transparency and replicability.

3.3 Conclusion and Link to Original Research Question: The conclusions are well-stated; however, there is a need to strengthen the connection to the original research question. While it is not necessary to repeat all analyzed variables, the conclusions should emphasize how the findings directly relate to the initial research question and propose practical implementations. This will enhance the overall impact of the study and provide clear guidance for future research or practical applications.

The article effectively presents statistically sound findings, but improvements can be made by reinforcing the link between the conclusions and the original research question. Additionally, the emphasis on job autonomy is commendable and aligns well with the existing literature on telecommuting. Consider framing the conclusions to better guide readers on practical implications and potential implementations based on the study's insights.

---

## Round 0.2 · Minor Revisions

Thank you for your thorough responses to the concerns raised by the reviewers. Although they are generally pleased with your clarifications, they have identified a few more minor issues that need your attention and further response.

**Language Note:** The review process has identified that the English language must be improved. PeerJ can provide language editing services - please contact us at copyediting@peerj.com for pricing (be sure to provide your manuscript number and title). Alternatively, you should make your own arrangements to improve the language quality and provide details in your response letter. – PeerJ Staff

Reviewer 1 ·

Basic reporting

- Minor grammar errors were detected. For example, in the result section, “The number of participants was 500 workers (mean 43.85 years, SD: 110.71, 250 males and 250 females) living in Japan.” should be "This study included 500 workers (mean 43.85 years, SD: 110.71, 250 males and 250 females) living in Japan." I suggest you have a colleague who is proficient in English and familiar with the subject matter review your manuscript, or contact a professional editing service.

- Please include p-values at the end of the sentence when comparing the demographics from the 336 samples included in the study and the 164 samples excluded from the study instead of showing t-statistics or z-statistics.

Experimental design

no comments

Validity of the findings

no comments

·

Basic reporting

Thank you for your attention to the suggestions in your paper and for the revision. However, some points have not yet been addressed and require further revision:

1) Abstract: There is a discrepancy between the objective outlined in the abstract and the background. While the abstract states that the study investigates the relationship between living environment factors and teleworkers' satisfaction, work engagement, productivity, and stress, the background mentions the aim of creating an optimum environment for telecommuting workers. Creating an optimum environment involves additional variables like work organization, relationships with supervisors, the nature of work, and distinctions between partial and total telework, which are not adequately addressed in this study. It is recommended to align the abstract and background to maintain coherence in the study's objectives.

2) Instead of adding a description regarding the consistency between before and after the coronavirus pandemic, the articles cited are from during the pandemic. Perhaps this systematic review can help you better describe the difference, paying attention to partial and total remote work:

Antunes ED, Bridi LRT, Santos M, Fischer FM. Part-time or full-time teleworking? A systematic review of the psychosocial risk factors of telework from home. Front Psychol. 2023 Feb 22;14:1065593. doi: 10.3389/fpsyg.2023.1065593.

Experimental design

The suggestions were addressed.

Validity of the findings

The suggestions were addressed.

---

## Round 0.3 · accepted · Accept

Thank you for addressing the reviewers' concerns.